# Iterative evolution of large-bodied hypercarnivory in canids benefits species but not clades

Mairin A. Balisi [1,2,3,4✉] & Blaire Van Valkenburgh [2]

Ecological specialization has costs and benefits at various scales: traits benefitting an individual may disadvantage its population, species or clade. In particular, large body size and hypercarnivory (diet over 70% meat) have evolved repeatedly in mammals; yet large hypercarnivores are thought to be trapped in a macroevolutionary "ratchet", marching unilaterally toward decline. Here, we weigh the impact of this specialization on extinction risk using the rich fossil record of North American canids (dogs). In two of three canid subfamilies over the past 40 million years, diversification of large-bodied hypercarnivores appears constrained at the clade level, biasing specialized lineages to extinction. However, despite shorter species durations, extinction rates of large hypercarnivores have been mostly similar to those of all other canids. Extinction was size- and carnivory-selective only at the end of the Pleistocene epoch 11,000 years ago, suggesting that large hypercarnivores were not disadvantaged at the species level before anthropogenic influence.

[1] La Brea Tar Pits and Museum, Los Angeles, CA 90036, USA. [2] Department of Ecology and Evolutionary Biology, University of California, Los Angeles, CA 90095, USA. [3] Department of Vertebrate Paleontology, Natural History Museum of Los Angeles County, Los Angeles, CA 90007, USA. [4] Department of Life and Environmental Sciences, University of California, Merced, CA 95343, USA. ✉email: mairin@g.ucla.edu

Modern mammal communities are depauperate in apex predators. Most regions, except Africa, harbor few coexisting large hypercarnivores (species ≥21 kg with diets that are >70% vertebrates). Extant North American ecosystems include at most two, the gray wolf and mountain lion, whereas late Pleistocene ecosystems such as Rancho La Brea included as many as five more: the extinct dire wolf, American lion, short-faced bear, and two saber-tooth cats[1]. Even more rare is bone-cracking, a modification of hypercarnivory that involves breaking open bones to obtain nutritious marrow; at present, the spotted and brown hyenas are our only extant bone-crackers[2]. The current paucity of large hypercarnivores encourages the perception that the costs of carnivory make diverse predator assemblages unsustainable[3–5]. The fossil record, however, preserves a richness of large-bodied mammalian hypercarnivores and bone-crackers, inviting inquiry into how these specializations may propagate through lineages and ecosystems, and why they are so rare today.

Extant large carnivorans tend to be hypercarnivores that consume prey as large as or larger than themselves[3,4]. This strategy is costly for several reasons. Large prey are less abundant and have patchier distributions than smaller prey, lengthening predators' search and pursuit times[6,7]; they are difficult to take down, making a carcass worth fighting over, which can lead to injury and death[8,9]; and they often are more dangerous than small prey[10,11]. Cracking large bones inflicts fracture and wear on teeth, the main tools of prey capture and food processing[12–14]. Juveniles must grow large enough to be able to learn attack behaviors and catch large prey, delaying acquisition of foraging skills[15,16]. On macroevolutionary scales, hypercarnivorous adaptations—e.g., loss of dental features to create slicing blades—are likely to be irreversible[17], limiting further morphological diversification.

Yet, hypercarnivory presents benefits that likely offset its costs. Barring predator specialization on a few prey species, meat is readily available[18]. It has high energy content[3] and is more efficiently digested than plants[19], permitting high basal metabolic rates, growth rates, and fecundity[18,20,21]. Hypercarnivores tend to be larger-bodied than non-hypercarnivorous relatives, affording greater dispersal ability across environments of variable resource availability[21,22]. Despite hypercarnivory's apparent irreversibility, it repeatedly punctuates the evolutionary history of the Carnivora[23], suggesting that it is successful[5].

The fossil record of North American dogs (Carnivora: Canidae) presents an ideal system for testing the impact of this specialization on the diversification of large predators. Fossil dogs arose in North America ~40 million years ago (Ma), radiating into over 130 species in three subfamilies: Hesperocyoninae, Borophaginae, and Caninae[24–26]. The extinct species surpass the extant species in ecomorphological range, including many large hypercarnivores[27]. Competitive interactions have been hypothesized among the subfamilies[28], and habitat changes over the Neogene may have precipitated morphological and behavioral shifts[29]; the roles of climate and competition in the rise and fall of canids remain an area of active study[30].

Negative relationships have emerged between dietary specialization (including hypercarnivory) and species duration in Hesperocyoninae, Borophaginae, and all Canidae[27,31], suggesting that specialization negatively impacts canid success. Here, we investigate further by (a) quantifying turnover rates at intervals to pinpoint differences in body mass and diet between extinction survivors and victims, (b) comparing rates of diversification (origination, extinction, and origination minus extinction, all relative to the total history of the group in question) between large hypercarnivores and all other canids as well as among subfamilies, and (c) tracking diversification rates in correlation with, first, traits as potential intrinsic drivers and, second, global

temperature estimated by oxygen isotopes as a possible extrinsic driver of canid diversification.

How have the costs and benefits of large-bodied hypercarnivory impacted extinction risk at the species level and, further, at the clade level? If being a large hypercarnivore increases the probability of going extinct, then we predict that, relative to smaller and/or less carnivorous canids, large hypercarnivores would have higher extinction frequencies, calculated as the proportion of species that become extinct relative to the total number of species in the interval. Alternatively, if the short-term benefits of large-bodied hypercarnivory outweigh the costs in the long-term, then large hypercarnivores may exhibit constant extinction rates on par with or lower than non-hypercarnivores.

In Hesperocyoninae and Borophaginae, the two canid subfamilies that are completely extinct, we find that diversification of large-bodied hypercarnivores appears constrained at the clade level, biasing specialized lineages to extinction. However, despite shorter species durations, extinction rates of large hypercarnivores have been mostly similar to those of all other canids. Extinction was size- and carnivory-selective only at the end of the Pleistocene epoch 11,000 years ago, suggesting that large hypercarnivores were not disadvantaged at the species level before anthropogenic influence.

## Results

**Large hypercarnivores**. The majority (100/132 analyzed species) of North American fossil canids over the last 40 Ma were <20 kg and fed on prey smaller than themselves (Fig. 1). Fossil canids spanned over an order of magnitude in mean size (from *Otarocyon cooki*, 1.67 kg; to *Epicyon haydeni*, 41.49 kg) and wide dietary range, including mesocarnivory and hypocarnivory[27]. Small- to medium-sized hypercarnivores exist—e.g., some extant foxes—but, because the energetic costs differ between smaller and larger hypercarnivores[4], we included these smaller species with all other canids. Based on our estimates of predator and prey body sizes, we categorized 32 species as large hypercarnivores, including *Enhydrocyon* (four species), *Ectopocynus simplicidens*, and *Osbornodon fricki* in Hesperocyoninae; *Aelurodon* (six species), *Paratomarctus euthos*, *Carpocyon webbi* and *robustus*, *Protepicyon* + *Epicyon* (three species), and *Borophagus* (eight species) in Borophaginae; and *Theriodictis? floridanus*, *Xenocyon* (two species) and two extinct species of *Canis* in Caninae.

**Survivor–victim analysis**. Figure 2 shows large hypercarnivorous species in the upper right quadrant (gray-shaded region) of each morphospace time slice. If being large and hypercarnivorous increases extinction risk, then, for each time slice, more extinct species (hollow shapes) would be in the gray-shaded region, and fewer extinctions representing smaller non-hypercarnivores would be in the unshaded region. Our analyses do not support this hypothesis. We found negligible differences in carnivory and body mass for most time intervals (Table 1). Particularly after correcting for multiple comparisons, the only significant difference between survivors and victims occurred at the end of the Pleistocene epoch (0.01 Ma; $P = 0.011$). At this time, three large hypercarnivores became extinct or were extirpated from North America—*Cuon alpinus*; *Canis armbrusteri*; and the dire wolf, *Canis dirus*—leaving behind a radiation of small and relatively hypocarnivorous foxes.

**Subfamily trends**. Despite minimal support for size- and carnivory-selective extinction, trends emerge within each subfamily. Starting 10 My after origin, Hesperocyoninae populated the large-hypercarnivore space (Fig. 2); the last surviving hesperocyonine, the large hypercarnivore *Osbornodon fricki*,

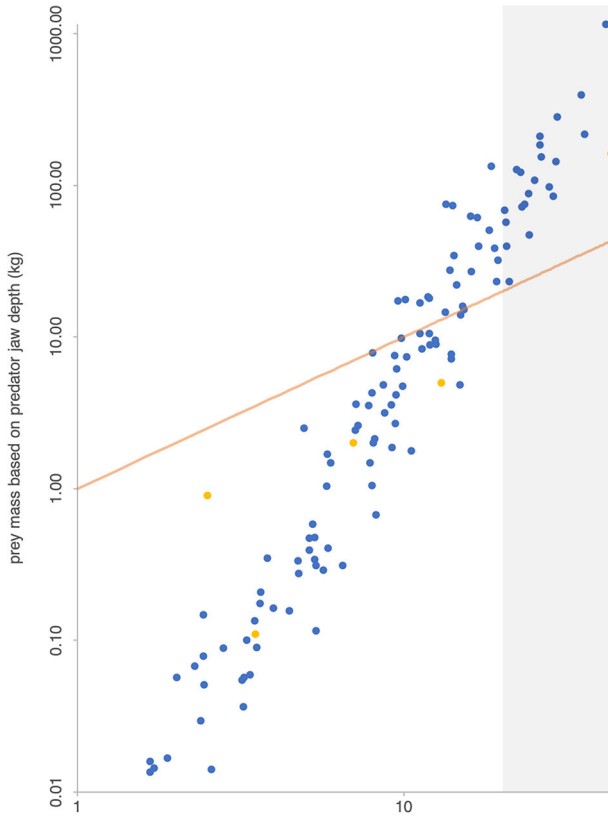

**Fig. 1 Plot of typical prey mass (kg) against canid mass (kg) for 130 species of extinct canids (blue points) and five species of extant canids (yellow points).** All plotted species are North American. Both *x* and *y* axes are log$_{10}$-transformed. Gray-shaded area marks estimated predator mass ≥20 kg. Orange line represents a 1:1 ratio, or prey mass equal to canid mass. Canid species falling above the orange line likely consumed prey larger than themselves, lending support to the categorization of "large hypercarnivore" based on size alone. Most North American fossil canids over the last 40 Ma were <20 kg and fed on prey smaller than themselves.

became extinct ~14.8 Ma. Borophagines boarded an evolutionary conveyor belt toward the niche left open by *O. fricki*'s extinction, eventually also becoming extinct ~2 Ma. Canines appear to follow this pattern most recently, until experiencing the only hypercarnivore-selective extinction ~0.01 Ma.

**Extinction dynamics of large hypercarnivores.** Starting with the hesperocyonine *Enhydrocyon* 30–27.9 Ma (Fig. 3), large hypercarnivores numbered fewer than all other canids for much of their history, except 13–8 Ma when their richness peaked (Figs. 2 and 3a). Large-hypercarnivore extinction started marginally higher than observed for all other canids but was surpassed by extinction of the latter at 20 Ma, although both rates are now statistically indistinguishable because of an increase in both large-hypercarnivore extinction rate and its credible interval starting ~7 Ma (Fig. 3b; Supplementary Table 1, Supplementary Fig. 1).

**Subfamily diversification rates.** For origination of Hesperocyoninae, the two-rate model showed highest support (Table 2), suggesting at least two origination rates in this subfamily; positive support for a rate decrease occurs at about 30 Ma (Fig. 4d–f). Borophagine origination rate, meanwhile, was nearly constant over the subfamily's duration (Fig. 4g–i, Table 2). Extinction of Hesperocyoninae accelerated at 29 Ma and again around 20 Ma;

for Borophaginae, extinction exceeded origination around 20 Ma. Different patterns characterize Caninae: origination and extinction rates have been approximately equal—producing zero to positive net diversification for much of North American canine history—and both rose gradually from 10 Ma to the present (Fig. 4j–l, Table 2).

**Correlation between rates and traits.** Origination and extinction rates do not correlate significantly with body mass, carnivory or a combination of the two traits in all Canidae or in any subfamily (Supplementary Table 2), although some rates appear to be weakly correlated (Supplementary Fig. 2).

**Correlation between rates and temperature.** Extinction rates for all canids, smaller non-hypercarnivores, and Caninae correlated positively with the oxygen-isotope record. Higher δ$^{18}$O values correspond to lower temperatures; therefore, there was higher turnover among species as global temperature decreased. All other correlations were not significant (Supplementary Fig. 3, Supplementary Table 3).

## Discussion
The costs and benefits of hypercarnivory are well known at the individual level[4,10,32]. At the species level, specialization may confer a short-term advantage by optimizing an organism for its environment and available resources; but, over long timescales, specialization can be an evolutionary trap for entire clades[23,33,34]. Hypercarnivorous adaptations that simplify the dentition to maximize the slicing surface—such as loss of cusps on teeth (e.g., canids with trenchant talonids) or loss of grinding teeth (e.g., feliforms)—exemplify Dollo's law: that a structure, once lost, is unlikely to be regained[17]. Relative to generalists bearing the ancestral condition, specialists with derived and reduced morphologies provide less material for evolvability and a narrower range of "next steps" for descendant species[35]. Consequently, on macroevolutionary timescales, canids appear repeatedly to board a conveyor belt toward progressively greater specialization, with few or no reversals[33]. Therefore, hypercarnivory evolving under Dollo's law inevitably will increase the relative frequency of hypercarnivory in the later history of a clade—until the clade vanishes, even if the possibility remains that the dietary shift is not the ultimate cause of extinction.

This evolutionary conveyor belt or macroevolutionary ratchet[31] is apparent in the survivor–victim analysis (Fig. 2). Each subfamily originates as small mesocarnivores, increasing in size and carnivory over time until—in the two extinct canid radiations—species enter and eventually vanish from the extreme quadrant of morphospace. Accordingly, clade extinction rates increase and eventually exceed origination rates after the rise of lineages leading to the first large hypercarnivores (*Enhydrocyon* in Hesperocyoninae ~29 Ma, followed by *Osbornodon* ~20 Ma; Borophagini in Borophaginae ~20 Ma) (Fig. 4). Specialization signals the beginning of clade decline.

Given this, and given the tendency of canid specialists toward shorter species durations (Fig. 2 in ref. [27])[32], one might expect higher extinction rates for large hypercarnivores, because extinction rate is the reciprocal of mean species longevity. This is not what we observed (Fig. 3; Supplementary Fig. 1, Table 1). The lack of evidence for higher extinction rates for large hypercarnivores (Fig. 3) might be due, at least partially, to how we binned the data in the analysis, comparing "large hypercarnivores" to "all other Canidae" in order to (1) focus not just on specialists but specifically on large hypercarnivores and (2) circumvent the problem of small sample sizes and poor fossil preservation for small hypocarnivores. Highly specialized species on the other end

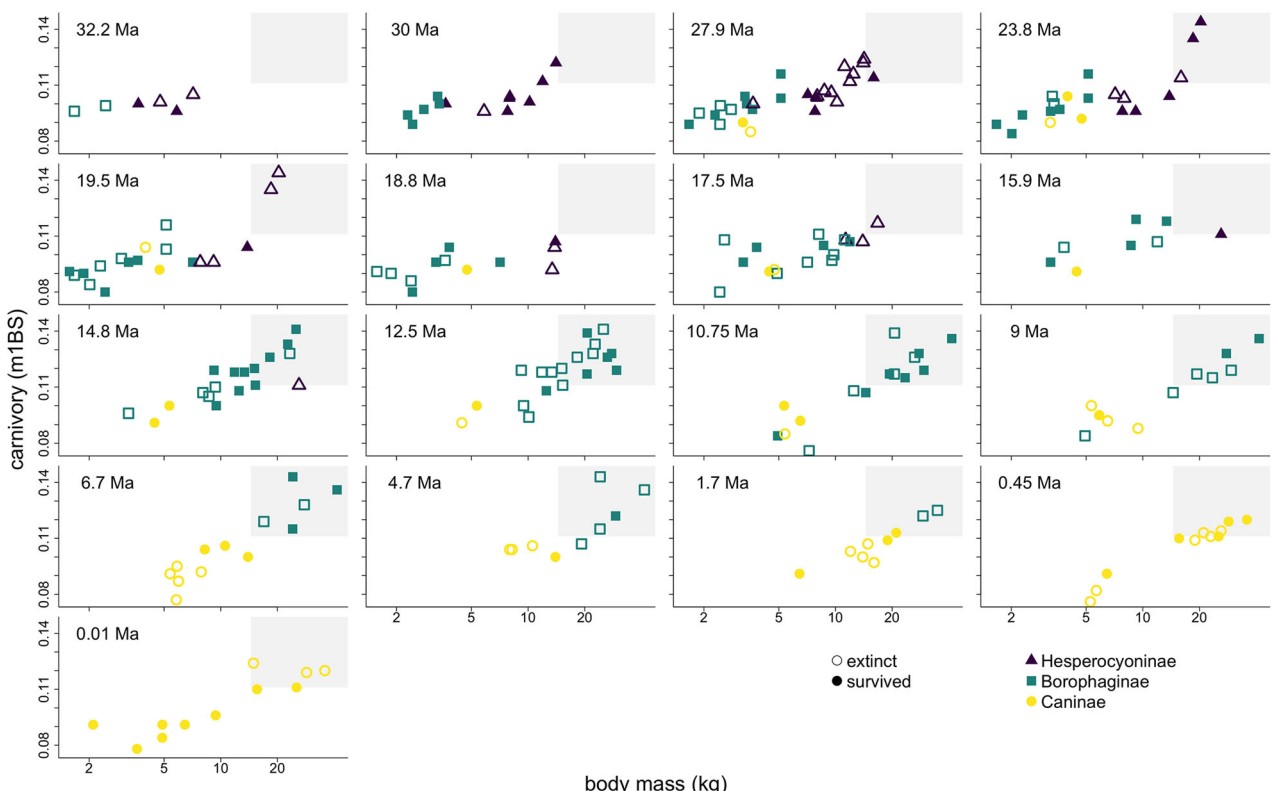

**Fig. 2 Ecomorphology of extinct and survived canid species over 17 time slices.** Gray boxes designate the large-hypercarnivore niche. Only the end-Pleistocene extinction, at 0.01 Ma, shows a significant difference in body mass and carnivory between extinct and survived.

| Table 1 Results of survivor–victim analysis over 17 time slices. | | | | | | | | |
|---|---|---|---|---|---|---|---|---|
| Time slice (millions of years ago) | A | | | B | | | C | | |
| | F | P | BH sig | Akaike weight, carnivory | Akaike weight, body mass | Akaike weight, carn:mass | $\chi^2$ | P | BH sig |
| 32.2 | 0.38 | 0.664 | No | **0.528** | 0.459 | 0.012 | NA | NA | NA |
| 30.0 | NA | NA | NA | **0.450** | 0.327 | 0.223 | NA | NA | NA |
| 27.9 | 1.454 | 0.227 | No | 0.426 | **0.478** | 0.096 | 0.027 | 0.864 | No |
| 23.8 | 0.27 | 0.698 | No | 0.475 | **0.476** | 0.050 | 0.014 | 0.919 | No |
| 19.5 | 1.581 | 0.215 | No | **0.768** | 0.182 | 0.050 | 1.626 | 0.303 | No |
| 18.8 | 0.77 | 0.415 | No | **0.516** | 0.476 | 0.008 | NA | NA | NA |
| 17.5 | 0.653 | 0.489 | No | 0.404 | **0.569** | 0.027 | 0.407 | 0.661 | No |
| 15.9 | 0.12 | 0.713 | No | 0.477 | **0.521** | 0.002 | 0.381 | 0.663 | No |
| 14.8 | 0.567 | 0.488 | No | **0.561** | 0.411 | 0.027 | 0.377 | 0.581 | No |
| 12.5 | 1.152 | 0.289 | No | 0.321 | **0.621** | 0.058 | 3.909 | 0.062 | No |
| 10.75 | 0.061 | 0.899 | No | 0.486 | **0.499** | 0.015 | 0.417 | 0.603 | No |
| 9.0 | 1.219 | 0.291 | No | **0.643** | 0.298 | 0.059 | 0.244 | 0.686 | No |
| 6.7 | 4.709 | 0.047 | No | 0.457 | **0.520** | 0.024 | 0.219 | 0.711 | No |
| 4.7 | 0.219 | 0.667 | No | **0.499** | 0.496 | 0.005 | 0.020 | 0.891 | No |
| 1.7 | 0.053 | 0.887 | No | 0.443 | **0.556** | 0.002 | 0.625 | 0.547 | No |
| 0.45 | 1.099 | 0.309 | No | **0.560** | 0.418 | 0.022 | 0.627 | 0.586 | No |
| 0.01 | **14.716** | **0.011** | **Yes** | **0.956** | 0.034 | 0.010 | 4.688 | 0.041 | No |

"BH sig" stands for significance after adjusting for multiple comparisons using the Benjamini–Hochberg (BH) method; results significant with the BH correction are in bold. For Akaike weights, the value for the best-supported model for each interval is in bold. (A) Results from permutational MANOVA indicating lack of significant differences between survived and extinct over most time slices. "NA" indicates no analysis possible because of only one survivor from the preceding time interval. (B) Akaike weights from logistic regressions with both body mass (log10mass) and carnivory (m1BS) coded as continuous variables. carn:mass is the interaction between carnivory and body mass. (C) Results of contingency tests with large-bodied hypercarnivory coded as a binary variable (yes/no). "NA" indicates no analysis possible because of a lack of large hypercarnivores in the preceding time interval.

of the spectrum—small hypocarnivores—also tend to have shorter durations[27] and therefore also should have higher extinction rates, hence potentially inflating the extinction rate of "all other Canidae". Nonetheless, neither size nor carnivory was correlated with diversification rates for all canids or any subfamily (Supplementary Table 2). Therefore, for most of canid history,

large-bodied hypercarnivory appears to have been a liability to clades, but not to species.

The lack of correlation between rates and traits presented here contrasts with recent work showing increased extinction rates in saber-toothed cats relative to other felids[36]. However, this difference makes sense considering that hypercarnivory in canids is

less extreme than that in felids. For example, while hypercarnivorous canids tend to have altered their lower molar tooth row to enhance slicing over grinding, they have retained the component teeth (m1–3). By contrast, saber-toothed cats tend to have lost the entire module of the dentition capable of chewing and grinding food: from the talonid basin of the lower carnassial to the post-carnassial molars (m2–3). By simplifying yet retaining structures, hypercarnivorous canids theoretically maintain a wider range of prey-consumption tools than did saber-toothed cats, which were more severely limited in their ability to process anything but meat. Many saber-toothed cats also had larger body sizes—some being the largest predators in their communities[37]—than

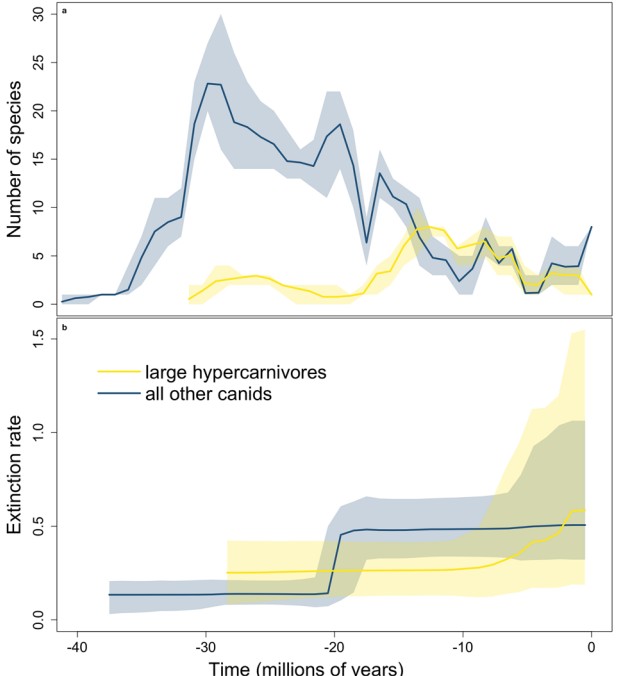

**Fig. 3 Diversification metrics of North American canid predators over time. a** Species richness and **b** extinction rates for North American fossil canids, both large hypercarnivores and not. Solid lines denote mean values; shading denotes 95% credible intervals. Large hypercarnivores numbered fewer than all other canids for much of their history. Large-hypercarnivore extinction started marginally higher than observed for all other canids but was surpassed by extinction of the latter at 20 Ma, although the two rates are now statistically indistinguishable because of an increase in both large-hypercarnivore extinction rate and its credible interval starting ~7 Ma. Credible intervals are based on 10,000,000 PyRate iterations.

contemporaneous hypercarnivorous canids, as seen in *Smilodon fatalis* and the dire wolf *Canis dirus* at the Pleistocene Rancho La Brea asphalt seeps[38]. The Rancho La Brea example introduces another difference between canids and felids that may play a role in extinction risk: hypercarnivorous canids tend to be pack hunters[39], while many extant felids tend to be solitary[40] (though *Smilodon* itself has been interpreted largely to have been social[41,42]). Among canids such as the African wild dog (*Lycaon pictus*), the ability to hunt in groups expands the range of prey sizes and enhances their ability to successfully defend their kills from theft[43]. These various considerations may explain the lack of increased risk in large hypercarnivorous canids, despite increased risk in hyper-specialized felids.

Apart from intrinsic ecomorphological constraints, overlapping diversity patterns suggest that inter-clade competition suppressed the evolution of large hypercarnivorous canids until later in canid history[30]. Amphicyonids (bear-dogs), felids, mustelids, nimravids and barbourofelids (false saber-toothed cats), procyonids, and ursids overlapped temporally with canids and also included large hypercarnivorous species[23]. Nimravids had become extinct and amphicyonids were declining by ~16 Ma, creating a "cat gap"[23] and leaving large-hypercarnivore niches open for canids alone until felids arrived in North America ~10 Ma[23,28]. However, overlapping diversity patterns alone are insufficient to infer competition and must be supported by evidence of similar ecomorphologies as a proxy for similar resource use. While ecomorphological overlap among these groups has been quantified[23,34], the resolution of canid taxonomy and phylogeny surpasses that of the other carnivoran clades, hindering precise ecomorphological comparison of canid species to non-canid carnivoran species. Taxon-free approaches (e.g., analyzing raw trait distributions not averaged by taxonomic units) might confirm the role of inter-clade competition in carnivoran diversification, particularly within a restricted context such as a single locality or paleocommunity.

Although competition may explain predator evolutionary divergence, climate can also influence predator evolution via bottom-up energy flow[44,45]. However, dynamics within the two extinct canid subfamilies, both of which increased in body size and carnivory before extinction, show no relationship with temperature (Supplementary Table 3; Supplementary Fig. 3). The subsets of the data that do show relationships with temperature (Caninae, Canidae, all other canids) largely comprise smaller non-predatory species. As well, Caninae is the only one of the three canid subfamilies to have radiated outside of North America[26], likely impacting its trait evolutionary dynamics by broadening the available niche spaces into which the subfamily could expand. Therefore, climate appears less important than

**Table 2 Relative probabilities of birth-death models with different numbers of rate shifts for all Canidae and each canid subfamily.**

| Model | Canidae | | Hesperocyoninae | | Borophaginae | | Caninae | |
|---|---|---|---|---|---|---|---|---|
| | Origination | Extinction | Origination | Extinction | Origination | Extinction | Origination | Extinction |
| 1-rate | 0 | 0 | 0.3109 | 0.1234 | **0.3986** | 0.0053 | 0.0252 | 0.0836 |
| 2-rate | 0.0278 | 0.2718 | **0.4096** | **0.4894** | 0.2978 | **0.7109** | **0.6548** | **0.7106** |
| 3-rate | 0.0613 | **0.5047** | 0.2085 | 0.2946 | 0.1639 | 0.2378 | 0.2524 | 0.1836 |
| 4-rate | **0.7044** | 0.1873 | 0.0589 | 0.0788 | 0.1159 | 0.0415 | 0.0605 | 0.0206 |
| 5-rate | 0.1794 | 0.0321 | 0.0107 | 0.0124 | 0.0216 | 0.0042 | 0.0068 | 0.0014 |
| 6-rate | 0.0246 | 0.0037 | 0.0014 | 0.0013 | 0.0021 | 0.0003 | 0.0004 | 0.0001 |
| 7-rate | 0.0024 | 0.0003 | 0.0001 | 0.0001 | 0.0001 | 0 | 0 | 0 |
| 8-rate | 0.0001 | 0 | 0 | 0 | 0 | 0 | 0 | 0 |
| 9-rate | 0 | 0 | 0 | 0 | 0 | 0 | 0 | 0 |

Bolded numbers are the highest probabilities in each column, signifying the most probable model for origination or extinction in each clade.

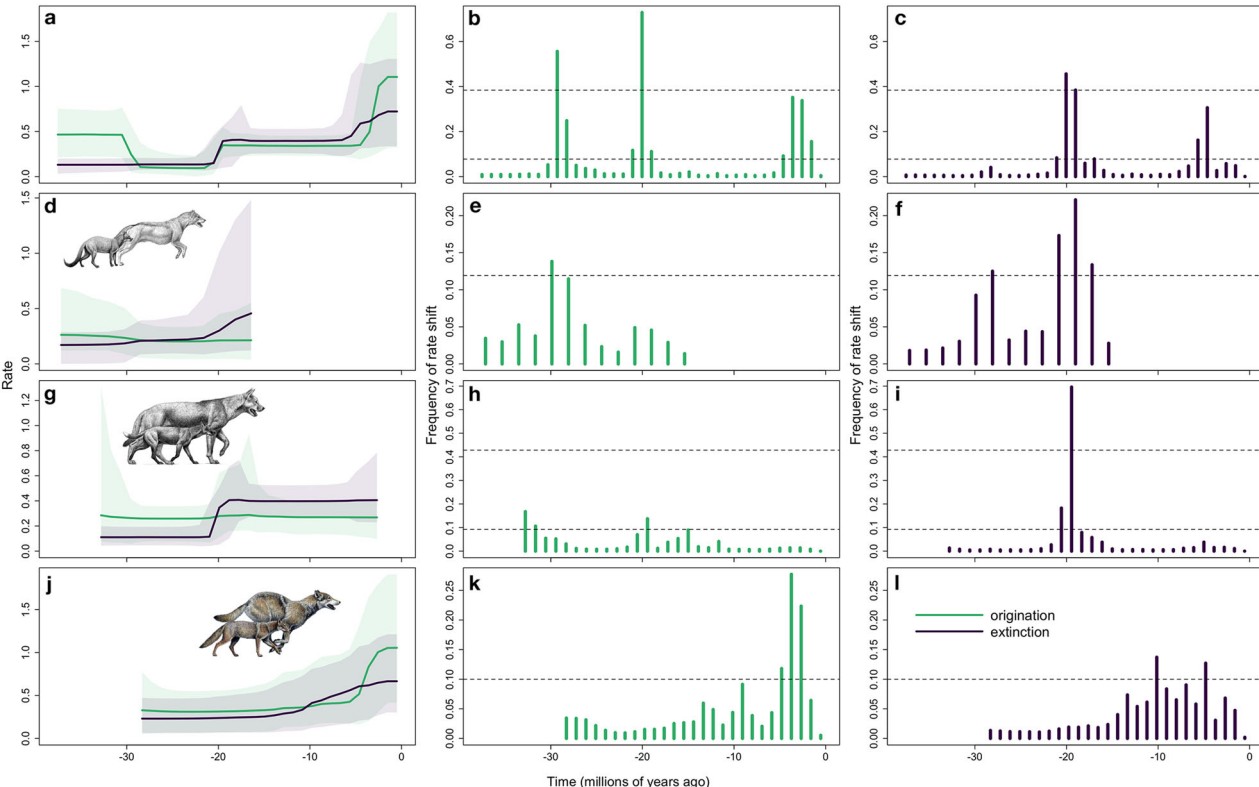

**Fig. 4 Origination and extinction rates through time with credible intervals, and histograms of inferred times of rate shifts for origination and extinction. a–c** All North American canids and the three subfamilies: **d–f** Hesperocyoninae, **g–i** Borophaginae, and **j–l** Caninae. The rate plots have different *y*-axis ranges to optimize visualization. Solid lines denote mean rates; shading denotes 95% credible intervals based on 10,000,000 PyRate iterations. Canid illustrations are by Mauricio Anton and used with permission. On the histograms, the horizontal dashed lines represent thresholds for positive evidence (bottom line) or strong evidence (upper line, when available) of a rate shift, as determined by Bayes Factors (bottom line, logBF = 2; upper line, logBF = 6; Hesperocyoninae and Caninae have only logBF = 2). Times of significant rate change—when significant posterior probability supports a rate shift—are indicated by bins in the histograms that show sampling frequencies for a rate shift exceeding the thresholds.

biotic interactions in the diversification of predator-dominated clades. Future work in this area would benefit from assessing regional environmental proxies alongside changes in faunal ecology and diversity (e.g., refs. [46,47]) as well as considering tectonic effects on diversification (e.g., refs. [48,49]).

Our results differ from those of Silvestro et al.[30], who first developed and tested the Python program PyRate, which we use here to quantify diversification in North American fossil canids. Silvestro et al. found strong support for two-rate models for both origination and extinction in the two extinct canid subfamilies: a decrease in origination rate and concomitant increase in extinction rate at ~26 Ma for Hesperocyoninae and 14 Ma for Borophaginae[30]. In comparison, our analyses similarly preferred two-rate models in Hesperocyoninae but with different timing of shifts (origination decreasing ~30 Ma and extinction increasing ~20 Ma) and identified a nearly constant origination rate eventually surpassed by extinction rate for Borophaginae ~20 Ma (Fig. 4, Table 2). This discrepancy likely stems from differences in both the analysis model and the source of the occurrence data. First, Silvestro et al.[30] modeled preservation as a non-homogeneous Poisson process (NHPP), allowing for rate heterogeneity only among lineages. In contrast, given the variability of the fossil record over time in terms of preservation quality and number of localities (Supplementary Figs. 4–6), we modeled preservation as a time-variable Poisson process (TPP) to incorporate both temporal and among-lineage variation in preservation rate. Second, Silvestro et al. drew fossil occurrences from the Paleobiology Database (PBDB). While we used some occurrences from the PBDB, we primarily drew from the Neogene Mammal

Mapping Portal, which records occurrence-specific versus interval-specific dates in the PBDB. Based on previous iterations of this study, interval-specific dates tend to have wider ranges than occurrence-specific dates, producing inflated estimates of stratigraphic range that are exacerbated in time periods with poor resolution (e.g., Arikareean North American Land Mammal Age).

No significant correlation emerged between diversification rates and ecomorphological traits across the entire history of canids. In part, this reflects our use of the Covar model in PyRate, which assumes a unidirectional relationship between a trait value and a diversification rate. In a previous study, we identified a nonlinear relationship between species duration and degree of carnivory: species at either end of the ecomorphological spectrum (large hypercarnivores and small hypocarnivores) tend to have shorter durations than mesocarnivores[27]. We analyzed these categories in the two-trait implementation of the Covar model, coding carnivory as a discrete trait while accounting for the effect of body mass coded as a continuous trait. The resulting lack of evidence for higher extinction rates for large hypercarnivores may stem from small hypocarnivores also having higher extinction rates, averaging out a potential relationship and causing Covar to find no trait-rate correlations significantly different from 0. As well, rather than being temporally explicit, Covar estimates an overall relationship through time without quantifying potential temporal effects, or how a correlation may change through time. This limitation may explain the apparent contradiction between the Covar results and those of the survivor–victim analysis—which divides time into snapshots, recovering ecomorphological selectivity in the most recent time slice—and why previous work

on North American fossil canids also has found largely no significant effect of ecomorphology on diversification rates[30]. Regardless, given that most intervals in the survivor–victim analysis display no signature of ecomorphological selectivity, the lack of significant association in the Covar analysis is expected. While nonlinear functions correlating rates to traits and temporal variation in dynamic models would be more complex and computationally intensive to implement, future analytical tools for tracking diversification rates and ecomorphological traits through time would benefit from developing these features.

The lack of size- or carnivory-selective extinction for all but the most recent period of North American canid history is unexpected given the rarity of extant large hypercarnivores and the shorter species durations of fossil hypercarnivores[27,31]. However, this result is consistent with recent studies of both extant and Pleistocene fauna that document a higher probability of extinction in the largest species, both on land and in the seas[50,51]. Moreover, this bias toward the removal of the largest species is unprecedented over the past 65 Ma of mammalian extinction events[51]. While these biodiversity changes began to take place relatively recently in North America, large carnivores started to decline millions of years earlier in Africa, where humans and their hominin ancestors have lived for much longer[52]. This temporal incongruity implies anthropogenic impacts—as opposed to factors concurrent between the two continents (e.g., changing forest cover)—in the long-term extinction of large carnivores[52]. While our data are restricted to a single diverse carnivoran family, our finding of minimal ecomorphological selectivity until the end-Pleistocene supports the idea that modern ecosystems are the product of, and continue to be subject to, trophic downgrading, a process that appears to be largely human-driven with negative effects on biodiversity and ecosystem resilience[1,53].

## Methods

**Measurement of species traits**. We measured carnassial blade length, jaw depth, and jaw length on specimens at the American Museum of Natural History (New York, NY), University of California Museum of Paleontology (Berkeley, CA), Los Angeles County Natural History Museum (Los Angeles, CA), Yale Peabody Museum (New Haven, CT), and John Day Fossil Beds National Monument (Kimberly, OR). When specimens were not easily accessible, we obtained measurements from the literature[24–26,54].

**Estimation of canid body size**. Because fragmentary fossils rarely preserve body size directly, we estimated size using Van Valkenburgh's regression on lower first molar length in extant canids[55]. Because carnivorans ≥20 kg have increased energetic costs and prey on larger species than do carnivorans <20 kg[3,4], we defined "large" size as species mean mass ≥20 kg.

**Estimation of prey body size**. A few species (e.g., some *Enhydrocyon*) yielded mass estimates just under 20 kg, even though other morphological evidence suggests that they were large and hypercarnivorous (e.g., ref. [56]). The regression estimates account only for mean mass; these species may have easily been over 20 kg in life. In addition, the mass regressions are based only on extant canids, all in subfamily Caninae. Hesperocyoninae and Borophaginae tend to have been built slightly more robustly than Caninae[56]; therefore, the estimates based on extant Caninae likely underestimate mass for the two extinct subfamilies. Given this, we supplemented the canid body mass estimates by estimating prey body size using a regression on jaw depth for extant canids[39]: $y = 5.583x - 6.482$, where $y$ is $\log_{10}$ prey size (kg) and $x$ is $\log_{10}$ jaw depth (mm) between the first and second lower molars. In this way, canid species estimated to have been just under 20 kg might still be categorized as "large hypercarnivores" based on an estimated prey size larger than themselves.

**Quantification of carnivory**. A suite of traits associated with increased bite forces and greater masticatory loads characterizes hypercarnivorous morphologies[57,58]. However, fragmentary fossils rarely preserve many of these traits. To maximize sample size, we quantified carnivory by a single metric: the length of the blade on the lower first molar (carnassial) relative to dentary length (m1BS). m1BS provides a more comprehensive quantification than other carnivory proxies, such as the blade length of the lower carnassial relative to the total lower carnassial length (RBL), because the calculation of m1BS relative to dentary length accounts for the shortening of the rostrum in some hypercarnivores (e.g., extant hyaenids) that aids

them in cracking bone[39]. Because not all species preserve intact dentaries, we estimated dentary length when needed using within-subfamily regressions based on the length of the lower first molar and calculated from fossil canid specimens with intact dentaries[27]. Quantifying carnivory as m1BS permitted inclusion of 127 fossil canid species in the morphometric sample. Five additional species were too fragmentary for inclusion in the morphometric sample but preserved enough material to be classified as small hypocarnivores; these five canids increased the sample to 132 species. Based on minimum values in extant large hypercarnivorous canids, we defined "hypercarnivory" in the fossil taxa as species with mean m1BS ≥0.107.

**Databases**. We compiled occurrences for North American fossil canids over the past 40 Ma from the Neogene Mammal Mapping Portal (NEOMAP, http://ucmp.berkeley.edu/neomap) and Paleobiology Database (PBDB, http://www.paleobiodb.org). NEOMAP links the Miocene Mammal Mapping Project[59] and the Quaternary Faunal Mapping Project[60], providing occurrences for published late Oligocene through Holocene mammals in the United States and many Quaternary localities in Canada. PBDB provides global occurrence data for organisms of all geologic ages. For the taxa and time periods of interest in this study, records of minimum and maximum locality age are more precise in NEOMAP than in PBDB; therefore, NEOMAP forms the bulk of the occurrences used here. PBDB was used for occurrences before the late Oligocene or absent from NEOMAP. Occurrences from both databases were repeatedly cross-checked for reliability against the published literature (e.g., refs. [24–26,61]). We compiled 3708 fossil occurrences for all Canidae: 314 Hesperocyoninae, 1265 Borophaginae, and 2129 Caninae.

**Statistics and reproducibility**. For survivor–victim analyses quantifying extinction selectivity per time interval, we examined 18 unequal-length time intervals over the past 40 My. The time intervals are biostratigraphic subdivisions of North American Land Mammal Ages (NALMAs) ranging from Orellan to Recent[62] (Supplementary Table 4). Each time slice compares the intervals before and after, totaling 17 slices. Species were categorized as winners (survivors) or losers (non-survivors), based on whether they were present in the subsequent interval. Lazarus taxa—taxa that disappear for one or more periods but later reappear—are included as having continuous records through the intervening interval/s in which they have no record. To determine whether or not survivors and non-survivors in a given time slice differ in traits, we pooled all taxa for each slice, sampled the pooled data with replacement 10,000 times, computed nonparametric test statistics using permutation (nonpartest() in package npmv[63]), and adjusted significance for multiple comparisons using the Benjamini–Hochberg method[64,65]. (While the more commonly used Bonferroni correction controls the familywise error rate by lowering the significance threshold for all tests, the less conservative BH method controls instead the false discovery rate (FDR), or the proportion of significant results that are false positives. We considered BH more appropriate than Bonferroni in this case given the relatively small number of tests and the preservational and temporal uncertainty inherent in the fossil record. We assigned FDR = 0.2, meaning that up to 20% of significant raw results are false positives.) We also examined three models for each slice using logistic regression: (a) extinction ~ body mass, (b) extinction ~ m1BS, and (c) extinction ~ body mass * m1BS, and quantified relative support for each model using Akaike weights (akaike.wts() in package paleoTS[66]) based on Akaike Information Criterion values corrected for small sample sizes (AICc() in package AICcmodavg[67]). In addition, given the sharp difference in energetic cost between large hypercarnivores and smaller carnivorans[3,4,32], we bootstrapped contingency tests (replicates = 10,000) with the independent variable as an ecomorphological category with two levels—large hypercarnivore, or not—and the dependent variable as extinction, also adjusting significance for multiple comparisons using the Benjamini–Hochberg method. These procedures were executed in R version 3.6.1[68].

To calculate diversification rates, we used the open-source Python program PyRate for joint estimation of species richness, preservation rates, and diversification rates (number of originations or extinctions per My over clade or group history)[30,69]. We partitioned the full dataset of all North American canid fossil occurrences into (1) large hypercarnivores and (2) all other canids. To account for uncertainty in the age of each occurrence, we generated 100 randomized sets of ages for each of the three datasets (all canids, large hypercarnivores, and canids excluding large hypercarnivores) by resampling the age of each occurrence uniformly within the respective temporal range. We then analyzed the datasets under a Reversible Jump Markov Chain Monte Carlo model with rate shifts (RJMCMC). We ran the analysis for the default of 10,000,000 RJMCMC iterations, sampled every 1000th iteration to obtain posterior estimates of the parameters, monitored effective sample sizes by visualizing the log files in Tracer[70], and discarded the first 200,000 iterations as burn-in. Because subsetting the taxa in this way assumes that only large hypercarnivores can give rise to large hypercarnivores—which is not true—we discarded the origination and net diversification rates and retained the extinction rates, which are determined only by the species itself. We also estimated origination, extinction, and net diversification rates for the three canid subfamilies separately and compared our results with those of Silvestro et al.[30].

PyRate's Covar model estimates the effect of a single continuous trait on diversification rates[69], yet our ecomorphology of interest—large-bodied hypercarnivory—is defined by two traits. Therefore, we implemented the Covar

model with two traits simultaneously. Following Piras et al.[36], we modified the Covar model to estimate the effect of carnivory (coded as a discrete trait: hyper-/meso-/hypocarnivore) on diversification rates while accounting for the effect of body mass (coded as a continuous trait), and ran the analysis using the -discrete and -twotrait flags in PyRate. In the Covar model generally, the parameters $\alpha_\lambda$ (correlation with origination rate) and $\alpha_\mu$ (correlation with extinction rate) are estimated from the data, quantifying the relationship between shifts in rates and in trait values. We ran the default of 10,000,000 iterations, sampled every 1000th, and discarded the first 2000 samples as burn-in. $\alpha > 0$ indicates a positive relationship between traits and rates; $\alpha < 0$ indicates a negative relationship. We considered the relationship significant if the distribution of 95% highest posterior densities of $\alpha$ did not overlap 0. We further used the original Covar model to test relationships between diversification rates and individual traits.

PyRate also models diversification rates changing through time as an exponential or linear function of a time-continuous correlate, such as temperature. We tested oxygen-isotope records published by Zachos et al.[71] for relationships with diversification. Higher $\delta^{18}O$ values signify lower temperature. We ran the default of 1,050,000 MCMC iterations, sampled every 1000th, and discarded the first 210,000 iterations as burn-in to obtain posterior estimates of the parameters $\gamma_\lambda$ (correlation with origination rate) and $\gamma_\mu$ (correlation with extinction rate). $\gamma > 0$ indicates positive correlation between diversification and temperature; $\gamma < 0$ indicates negative correlation. We considered the relationship significant if the distribution of 95% highest posterior densities of $\gamma$ did not overlap 0.

**Reporting summary**. Further information on research design is available in the Nature Research Reporting Summary linked to this article.

## Data availability
The morphometric, occurrence, and PyRate datasets generated and analyzed in this study are available on the Dryad repository: https://doi.org/10.6071/M3M08P[72].

## Code availability
The R code underlying the present analyses and figures are available on the Dryad repository: https://doi.org/10.6071/M3M08P[72].

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

## Acknowledgements
We wish to thank for our collections work: J. Galkin and J. Flynn (American Museum of Natural History (AMNH), New York, NY); M. Fox and D. Brinkman (Yale Peabody Museum, New Haven, CT); P. Holroyd (University of California Museum of Paleontology, Berkeley, CA); J. Samuels and C. Schierup (John Day Fossil Beds National Monument, Kimberly, OR); and S. McLeod and V. Rhue (Natural History Museum of Los Angeles County (LACM), CA). The Macroevolution in R (National Center for Ecological Analysis and Synthesis, Santa Barbara, CA) and Fossilworks workshops (Macquarie University, Sydney, Australia), as well as discussions with X. Wang, J. Chang, D. Silvestro, M. Juhn, and J. Payne, helped this project. Fellowships from UCLA Ecology and Evolutionary Biology and Graduate Division funded M.A.B. Funding from the AMNH (Theodore Roosevelt Memorial Grant), the LACM, and the U.S. National Science Foundation (DEB-1501931; DBI-1812301) supported this research.

## Author contributions
M.A.B. conceptualized the study, curated the data, conducted analysis, acquired funding, formulated methodology, administered the project, visualized the results, wrote the original draft, and reviewed and edited the paper. B.V.V. conceptualized the study, provided resources, supervised the project, and reviewed and edited the paper.

## Competing interests
The authors declare no competing interests.
