## [Peer Review File · Communications Biology]

Reviewers' comments:

Reviewer #1 (Remarks to the Author):

The ms was a pleasure to read. It is straight and simple, very credible and clear. I believe this is ideal for Communications Biology format. I believe there is a couple of citations missing which are very relevant to the ms, in particular, one showing increased extinction rates in sabertoothed cats as compared to other felids. Concerning this very paper, it is worth considering, in the manuscript, that sabertooth, contrary to hypercarnivorous canids, bear no teeth at all behind lower carnassials. In addition, there is no carnivore in the field capable to outweigh a sabertooth. These couple of simple considerations could explain why finding increased extinction risk in hypercarnivorous felids (sabertooths) is more likely than in hypercarnivorous canids. I believe 2-3 lines explaining these differences are very much worth adding. I very much enjoyed the conveyor belt hypothesis description. This is one among a few studies now recognizing how common this pattern is in the record, and how much correct Cope was (if it were not for the process behind it) in formulating his law of unspecialized.

All in all, I think this is a great piece and recommend publication pending the few worthy additions I've highlighted above

Regards

Reviewer #2 (Remarks to the Author):

This is very interesting paper that investigates the effect of body size and dietary specialization on extinction regimes in North American Canidae. The topic has been studied before, but previous results either found weak evidence or no evidence that body size and/or dietary specialization might be relevant. One main difference in the current study is that it investigates this potential association in different time intervals, one at a time. The results suggest that extinction selectivity only happened in two specific time intervals, including a higher extinction for large hypercarnivores in the end of the Pleistocene. I think the paper is interesting and has potential but there are a few points (most likely to be easy to address) that need to be addressed before it is ready for publication.

1) I suspect that breaking the data into large carnivores and other-mammals is not a valid procedure when considering the assumptions of the birth-death model. This might be ok for extinction dynamics given that extinction is only determined by the species itself, but not for speciation where it matters from where the new species it came from. For example, by splitting the data into large carnivores, the model is assuming that only large carnivores can give "birth" to a new large carnivore. I might be wrong here, but I think this is not a valid procedure and in fact the paper does not gain much by adding those analysis that split the data according to these criteria. There were 3 sets of analysis that looked into the same problem, and all converge to similar results, but I think that just having the covar model for the monophyletic clades and the survivor-victim analysis should suffice. Hence my suggestion is to remove the PyRate analysis done by partitioning the data into large hypercarnivores vs other canids, and keep only the ones that analyzed the covar model within monophyletic clades, as well as the survival/victim analysis.

2) Not clear how the covar model was implemented with two traits simultaneously. I looked into the Pyrate github and could not find any explanation. In fact, it is said in the methods section that the covar model was modified. Given that it is not a usual procedure, the authors need to be better explained how that analysis was done.

3) Results and discussion on Hesperocyoninae suggest no changes in speciation dynamics but I think the analysis done in the paper suggests the opposite. Table S2 shows evidence for at least 2 rates of speciation in Hesperocyoninae. Infact the two-rate model is the one with the higher support. Figure S3 also strongly suggests a speciation shift (BF higher than 6) at about 30 My. I suspect the visual impression given by figure 3 (and to a less extent supplemental figures) is driven by the plotting options used, in particular the wide range of values for the y-axis which compresses the variation in speciation. I would change the range of the y-axis for each panel to make it easier to see the dynamics. I understand the reasoning to make them similar, but the authors can make a note on caption to remind the different values to the reader.

4) I think (and this might have been said in the text, sorry if I missed) the result from the covar model is non-significant because it estimates an overall relationship, where no temporal effect is considered. Hence given that for most intervals there is no signature in the survivor-victim analysis it is not surprising to not find a significant association in this analysis. It might be interesting to add a bit more discussion on this topic, to remind the reader that the covar model is not temporally explicit. This discussion could also help to serve two purposes: explicitly state that the apparent contradiction (which might be in the mind of a given reader) between the covar and survivor/victim analysis might not be a contradiction after all; discuss why previous work has not found a significant effect. It also exposes some limitations of the covar model, which might encourage new developments of the method.

5) Given that the curatorial work done by the authors likely rendered their dataset an improved version of was done before, I think it would be good to include this dataset in the available data. I downloaded the data available and, if I am not wrong, the raw data was not there. Even though people could re-download the data from PBDB and MIOMAP, there seems to be some important curatorial work done by the authors that would be nice to be shared with the whole community. This is a big contribution of the paper.

6) Given the large number of independent statistical tests for the survivor-victim analysis and the fact that some species might be present in more than one time slice (hence in multiple tests), I wonder if there should not be a correction for multiple statistical tests. I was not sure...

Reviewer #3 (Remarks to the Author):

I read with interest this manuscript, which I found well written and think pushes forward our understanding of the canid clade evolution. The authors analyze the fossil record of the three subfamilies of North American dogs and analyze the effect of dietary specialization toward hyper-carnivory, body size evolution and climate change on the longevity and extinction of species. They find that hyper-carnivory has evolved independently multiple times and, while it does not increase species-level extinction, it is associated with extinctions of clades. Another finding that I found really interesting is that increased extinction rates in hyper-carnivores is instead associated with anthropogenic effects.

I would like the authors to clarify the link between species longevity and carnivory.

If the species duration in large hypercarnivores is shorter than their extinction rates **should** be

higher, because the extinction rate is the reciprocal of the mean species longevity. I suggest adding a plot of estimated longevities across all species to show how strong this difference in longevity is among ecotypes.

On p. 7 the authors identify what could be the key reason for the discrepancy between the apparent evolutionary-trap effect of hypercarnivory and the lack of differences in extinction rates. If hypercarnivory evolves under Dollo's law, it will inevitably lead to a relative increase in frequency of hypercarnivory throughout the history of a clade. Thus, when the clade goes extinct (perhaps for reasons unrelated to diet) we will see that the clade has evolved towards hypercarnivory, even if this diet change is not per se linked with their extinction. I think this possibility should be discussed more prominently in the paper.

Another aspect that I think the authors could elaborate a bit more upon is the comparison with body size and specialization of felids, which also evolved multiple times with associated changes in extinction (e.g. <https://doi.org/10.1016/j.palaeo.2018.01.034>)

Finally, it might be worth mentioning that Caninae spread outside of North America. Does that have consequences on the evolutionary dynamics of hyper-carnivory and body size increase?

Minor points

The 0.047 P-value reported on p. 5 might actually not even be significant, if you correct the significance threshold for multiple testing (e.g. Bonferroni correction).

P. 6: "net diversification (origination – extinction) never differed significantly from 0" could this also reflect uncertainties due to limited sample size?

P. 6: "confidence interval" should actually read "credible interval" since it's a posterior distribution.

p. 6: "origination and extinction rates have been *approximately* equal"

P. 6 and Table S3: Some of the traits seems to be at least weakly significant. Maybe another way to display the results is to calculate the posterior probability that e.g. $\alpha_{\lambda} > 0$. This is simply the frequency at which $\alpha_{\lambda} > 0$ in the posterior samples.

Reviewers' comments:

Reviewer #1 (Remarks to the Author):

The ms was a pleasure to read. It is straight and simple, very credible and clear. I believe this is ideal for Communications Biology format. I believe there is a couple of citations missing which are very relevant to the ms, in particular, one showing increased extinction rates in sabertoothed cats as compared to other felids. Concerning this very paper, it is worth considering, in the manuscript, that sabertooth, contrary to hypercarnivorous canids, bear no teeth at all behind lower carnassials. In addition, there is no carnivore in the field capable to outweigh a sabertooth. These couple of simple considerations could explain why finding increased extinction risk in hypercarnivorous felids (sabertooths) is more likely than in hypercarnivorous canids. I believe 2-3 lines explaining these differences are very much worth adding. I very much enjoyed the conveyor belt hypothesis description. This is one among a few studies now recognizing how common this pattern is in the record, and how much correct Cope was (if it were not for the process behind it) in formulating his law of unspecialized.

All in all, I think this is a great piece and recommend publication pending the few worthy additions I've highlighted above

Regards

- Many thanks to this Reviewer and Reviewer #3 for suggesting the Piras *et al.* saber-toothed cat paper. We had referred to this paper while writing our study but amazingly neglected to cite it. We have corrected our omission and added a discussion contrasting canid hypercarnivory vs saber-toothed morphology.

Reviewer #2 (Remarks to the Author):

This is very interesting paper that investigates the effect of body size and dietary specialization on extinction regimes in North American Canidae. The topic has been studied before, but previous results either found weak evidence or no evidence that body size and/or dietary specialization might be relevant. One main difference in the current study is that it investigates this potential association in different time intervals, one at a time. The results suggest that extinction selectivity only happened in two specific time intervals, including a higher extinction for large hypercarnivores in the end of the Pleistocene. I think the paper is interesting and has potential but there are a few points (most likely to be easy to address) that need to be addressed before it is ready for publication.

1) I suspect that breaking the data into large carnivores and other-mammals is not a valid procedure when considering the assumptions of the birth-death model. This might

be ok for extinction dynamics given that extinction is only determined by the species itself, but not for speciation where it matters from where the new species it came from. For example, by splitting the data into large carnivores, the model is assuming that only large carnivores can give “birth” to a new large carnivore. I might be wrong here, but I think this is not a valid procedure and in fact the paper does not gain much by adding those analysis that split the data according to these criteria. There were 3 sets of analysis that looked into the same problem, and all converge to similar results, but I think that just having the covar model for the monophyletic clades and the survivor-victim analysis should suffice. Hence my suggestion is to remove the PyRate analysis done by partitioning the data into large hypercarnivores vs other canids, and keep only the ones that analyzed the covar model within monophyletic clades, as well as the survival/victim analysis.

- This comment is an excellent and valid critique that had not occurred to us, and we sincerely thank the reviewer for pointing it out. In response, we have removed most of the PyRate analysis (origination rates and net diversification rates) that was done by partitioning the data into large carnivores versus other canids. However, we have retained the extinction dynamics data from this analysis, because (a) as the reviewer notes, extinction is determined only based on the species itself, and (b) especially given the low sample sizes resulting from subdividing Canidae into time intervals, a continuous-time Bayesian analysis supplementing the discrete-time frequentist survivor-victim analysis would be useful. We have added sentences explaining this to the relevant sections.

2) Not clear how the covar model was implemented with two traits simultaneously. I looked into the Pyrate github and could not find any explanation. In fact, it is said in the methods section that the covar model was modified. Given that it is not a usual procedure, the authors need to be better explained how that analysis was done.

- Thank you. We added wording to the relevant Methods section to clarify (1) what the original Covar model does, (2) why we modified the model, (3) how the modified model differs, (4) the PyRate flags that we used to run the analysis, and (5) that this modification was not original to us. Rather, we followed Piras and Silvestro *et al.* (2018, *Palaeogeography, Palaeoclimatology, Palaeoecology*) in implementing this procedure, and now we have credited them for doing so (apologies for the omission).

3) Results and discussion on Hesperocyoninae suggest no changes in speciation dynamics but I think the analysis done in the paper suggests the opposite. Table S2 shows evidence for at least 2 rates of speciation in Hesperocyoninae. Infact the two-rate model is the one with the higher support. Figure S3 also strongly suggests a speciation shift (BF higher than 6) at about 30 My. I suspect the visual impression given by figure 3 (and to a less extent supplemental figures) is driven by the plotting options used, in particular the wide range of values for the y-axis which compresses the variation in

speciation. I would change the range of the y-axis for each panel to make it easier to see the dynamics. I understand the reasoning to make them similar, but the authors can make a note on caption to remind the different values to the reader.

- y-axes changed in Figures 3, S2, and S3, and captions updated accordingly. We also have clarified the relevant text regarding two probable origination rates in Hesperocyoninae.

4) I think (and this might have been said in the text, sorry if I missed) the result from the covar model is non-significant because it estimates an overall relationship, where no temporal effect is considered. Hence given that for most intervals there is no signature in the survivor-victim analysis it is not surprising to not find a significant association in this analysis. It might be interesting to add a bit more discussion on this topic, to remind the reader that the covar model is not temporally explicit. This discussion could also help to serve two purposes: explicitly state that the apparent contradiction (which might be in the mind of a given reader) between the covar and survivor/victim analysis might not be a contradiction after all; discuss why previous work has not found a significant effect. It also exposes some limitations of the covar model, which might encourage new developments of the method.

- We had considered this in trying to resolve the “contradiction” but hesitated to discuss what we interpreted as a methodological issue. Thank you for validating our thoughts. We have added a paragraph discussing this.

5) Given that the curatorial work done by the authors likely rendered their dataset an improved version of was done before, I think it would be good to include this dataset in the available data. I downloaded the data available and, if I am not wrong, the raw data was not there. Even though people could re-download the data from PBDB and MIOMAP, there seems to be some important curatorial work done by the authors that would be nice to be shared with the whole community. This is a big contribution of the paper.

- Added the curated occurrences dataset as a third supplement on Dryad.

6) Given the large number of independent statistical tests for the survivor-victim analysis and the fact that some species might be present in more than one time slice (hence in multiple tests), I wonder if there should not be a correction for multiple statistical tests. I was not sure...

- Reviewer #3 also commented on the possible need to correct for multiple comparisons, validating this reviewer’s comment. In response, we adjusted the significance level using the Benjamini-Hochberg method. More details are in our response to Reviewer #3 below.

Reviewer #3 (Remarks to the Author):

I read with interest this manuscript, which I found well written and think pushes forward our understanding of the canid clade evolution. The authors analyze the fossil record of the three subfamilies of North American dogs and analyze the effect of dietary specialization toward hyper-carnivory, body size evolution and climate change on the longevity and extinction of species. They find that hyper-carnivory has evolved independently multiple times and, while it does not increase species-level extinction, it is associated with extinctions of clades. Another finding that I found really interesting is that increased extinction rates in hyper-carnivores is instead associated with anthropogenic effects.

*I would like the authors to clarify the link between species longevity and carnivory. If the species duration in large hypercarnivores is shorter than their extinction rates *should* be higher, because the extinction rate is the reciprocal of the mean species longevity. I suggest adding a plot of estimated longevity across all species to show how strong this difference in longevity is among ecotypes.*

- We have such a figure—estimated longevity across all species plotted against ecomorphology—in an earlier publication (Figure 2 in Balisi, Casey, and Van Valkenburgh, 2018, *Royal Society Open Science*). To make this reference explicit, we have added a citation to the figure itself on page 7.

On p. 7 the authors identify what could be the key reason for the discrepancy between the apparent evolutionary-trap effect of hypercarnivory and the lack of differences in extinction rates. If hypercarnivory evolves under Dollo's law, it will inevitably lead to a relative increase in frequency of hypercarnivory throughout the history of a clade. Thus, when the clade goes extinct (perhaps for reasons unrelated to diet) we will see that the clade has evolved towards hypercarnivory, even if this diet change is not per se linked with their extinction. I think this possibility should be discussed more prominently in the paper.

- Added a sentence discussing this possibility to the end of the first Discussion paragraph, on page 7.

*Another aspect that I think the authors could elaborate a bit more upon is the comparison with body size and specialization of felids, which also evolved multiple times with associated changes in extinction
(e.g. <https://doi.org/10.1016/j.palaeo.2018.01.034>)*

- Added a relevant discussion paragraph in response to this and Reviewer #1.

Finally, it might be worth mentioning that Caninae spread outside of North America. Does that have consequences on the evolutionary dynamics of hyper-carnivory and

body size increase?

- Added a sentence on page 8 to address this briefly.

Minor points

The 0.047 P-value reported on p. 5 might actually not even be significant, if you correct the significance threshold for multiple testing (e.g. Bonferroni correction).

- In response to Reviewer #2, we adjusted for multiple comparisons using the Benjamini-Hochberg (BH) procedure, which is less conservative than the Bonferroni correction. The more commonly used Bonferroni correction controls the familywise error rate by lowering the alpha value for all tests; meanwhile, BH controls the false discovery rate (FDR), or the proportion of significant results that are false positives. We considered the BH procedure more appropriate than the Bonferroni method given the relatively small number of tests and the preservational and temporal uncertainty inherent in the fossil record. Indeed, under BH, the raw *p*-value of 0.047 is not significant.

P. 6: “net diversification (origination – extinction) never differed significantly from 0” could this also reflect uncertainties due to limited sample size?

- Agreed. However, we removed the relevant sentences in response to a critique from a reviewer above regarding the validity of calculating origination rates for large hypercarnivores vs not, so this comment no longer applies.

P. 6: “confidence interval” should actually read “credible interval” since it’s a posterior distribution.

- Corrected.

*p. 6: “origination and extinction rates have been *approximately* equal”*

- Corrected.

P. 6 and Table S3: Some of the traits seems to be at least weakly significant. Maybe another way to display the results is to calculate the posterior probability that e.g. $\alpha_{\lambda} > 0$. This is simply the frequency at which $\alpha_{\lambda} > 0$ in the posterior samples.

- We agree with the importance of displaying weakly significant correlations and have created a new figure (Figure S4): a series of histograms that illustrates more transparently the results presented in Table S3.

REVIEWERS' COMMENTS:

Reviewer #2 (Remarks to the Author):

This is an improved version of a very interesting paper I reviewed previously. The authors have addressed all my concerns and I think the paper is almost ready for publication. It will be a very nice contribution and I have only minor suggestions. On reading it again I thought that one specific point might still need a bit of work, which I think might be mostly done by wording and better discussing some possible interpretations of some of the results. In particular I am still curious by the apparent contradiction (in fact highlighted by the other reviewer in the last round of reviews) showing smaller longevities but not higher extinction rate for hypercarnivores. In my first reading of the paper I thought this was produced by some limitations of the methods used (which the authors now more explicitly talk about), but I think there might be more to it. Below I present this point and a few other that are really minor.

1) I have two observations regarding the relationship between species longevities and carnivory in the light of a lack of evidence for higher extinction rate for hypercarnivores. As explicitly pointed out by the other reviewer, smaller longevities should reflect higher extinction rates, but the authors argued that for hypercarnivory, extinction rate is not higher, but still suggest they have smaller longevities by citing previous work. They present some potential explanations, but I think there might be more to it. In fact, the author's response brought me to the very interesting paper by Basili et al 2018 (great work!). I looked at the figure 2 of that paper and a few ideas came to mind. From Balisi et al 2018: "We find a nonlinear relationship between species duration and degree of carnivory: species at either end of the carnivory spectrum tend to have shorter durations than mesocarnivores." From figure 2 of that paper there seems to be lack of long-lived species for species with higher hypercarnivory index, this is particularly striking for Borophaginae, except for one species. When looking at the figure there is a "prohibited upper right triangle" in that scatter plot (again specially for Borophaginae), which indeed suggests smaller longevities for larger hypercarnivores. The potentially strange result is that this should produce higher extinction rates as well. On the other hand, the Balisi et al 2018 paper suggests that specialization per se results in lower extinction, not only hypercarnivory. Hence:

A- It might be the case that the lack of evidence for higher extinction rates for large hypercarnivores in the current paper (which compares it to "other Canidae"; figure 2) is, at least partially, due to the fact that highly specialized species in the other end of the spectrum (small hypocarnivores) also have higher extinction rates, and are hence "inflating" the extinction rate of "other Canidae". This could be discussed, including the possibility that breaking up the data as it was done here (hypercarnivorous vs other Canidae) might have driven the PyRate analysis shown on figure 2 which does not find evidence for higher extinction rates in large hypercarnivores when compared to "other Canidae". The result that hypercarnivore does not lead to higher extinction rate when compared to the other Canidae is still valid, but this is partially due to what are the "other Canidae". Clarifying that might be helpful for the reader.

B- I wonder if the covar-model would not also be influenced by the result presented by Balisi et al 2018. In the 2018 paper the authors "find a nonlinear relationship between species duration and degree of carnivory: species at either end of the carnivory spectrum tend to have shorter durations than mesocarnivores." Does the covar model assume a "unidirectional" relationship between a trait value and rate? If it does, then a lack of evidence here for higher extinction rates for hypercarnivores might also be due to the fact that species at the other side of the spectrum have higher extinction rate and PyRate cannot find a slope different than 0.

If the two points above make sense, then I see two ways to deal with it. The first is simply to acknowledge in the text that those are potential methodological limitations and choices and bring the Balisi et al 2018 results into the discussion. The other is to try to re-run the analysis using three categories for the analysis shown in figure 2, and a non-linear function to relate rate to trait value for the covar model. The second option might be difficult if it is hard to implement the covar modifications, and if splitting the data into three categories diminishes the sample size to values that become too small to analyze. That said I think simply discussing might suffice for now. The paper already represents a nice advance in our understanding of canid evolution, and future work might try to deal with these methodological issues.

2) In my previous review I pointed out that Hesperocyoninae in fact showed more evidence for more than one speciation rate (see table S2) and although the authors have mentioned that in the results it seems that the discussion still considers Hesperocyoninae to have constant speciation rate (see lines 241 to 244). A two-rate model for speciation is not only the preferred model, but if we compare a single rate model it only has 0.31 support against 0.69 for the other models which suggest 2 or more rates. To my mind this is strong evidence that your analysis does suggest speciation is not constant through time for Hesperocyoninae. It might even have two shifts (although one shift is the preferred model). Hence the comparison to previous results might be revisited here. It seems that the results for Borophaginae is where you find strong support for differences when compared to previous work and the discussion on why this is the case is already there and a very nice one by the way.

3) Although the authors refer to another paper when mentioning the estimates of prey size, it might be a good idea to briefly explain how this is done in this paper. To not break the paper flow this could be done in Figure S1 caption for example. Just a suggestion.

Reviewer #3 (Remarks to the Author):

Thank you for revising your manuscript and for addressing the points raised in my previous review. I think the paper reads really nicely and provides a substantial contribution to our understanding of the evolution of carnivores. The anthropogenic impact on the extinction of large carnivores in Africa has been discussed in a recent paper (<https://onlinelibrary.wiley.com/doi/full/10.1111/ele.13451>). It's really interesting to see the temporal difference between anthropogenic effects in Africa, where humans and their ancestors have been around for much longer compared to a recently colonized continent, maybe worth a note in the paper?

REVIEWERS' COMMENTS:

Reviewer #2 (Remarks to the Author):

This is an improved version of a very interesting paper I reviewed previously. The authors have addressed all my concerns and I think the paper is almost ready for publication. It will be a very nice contribution and I have only minor suggestions. On reading it again I thought that one specific point might still need a bit of work, which I think might be mostly done by wording and better discussing some possible interpretations of some of the results. In particular I am still curious by the apparent contradiction (in fact highlighted by the other reviewer in the last round of reviews) showing smaller longevities but not higher extinction rate for hypercarnivores. In my first reading of the paper I thought this was produced by some limitations of the methods used (which the authors now more explicitly talk about), but I think there might be more to it. Below I present this point and a few other that are really minor.

- Thank you for the kind words, and for your comprehensive reading of our revisions.

1) I have two observations regarding the relationship between species longevities and carnivory in the light of a lack of evidence for higher extinction rate for hypercarnivores. As explicitly pointed out by the other reviewer, smaller longevities should reflect higher extinction rates, but the authors argued that for hypercarnivory, extinction rate is not higher, but still suggest they have smaller longevities by citing previous work. They present some potential explanations, but I think there might be more to it. In fact, the author's response brought me to the very interesting paper by Basili et al 2018 (great work!). I looked at the figure 2 of that paper and I few ideas came to mind. From Balisi et al 2018: "We find a nonlinear relationship between species duration and degree of carnivory: species at either end of the carnivory spectrum tend to have shorter durations than mesocarnivores." From figure 2 of that paper there seems to be lack of long-lived species for species with higher hypercarnivory index, this is particularly striking for Borophaginae, except for one species. When looking at the figure there is a "prohibited upper right triangle" in that scatter plot (again specially for Borophaginae), which indeed suggests smaller longevities for larger hypercarnivores. The potentially strange result is that this should produce higher extinction rates as well. On the other hand, the Balisi et al 2018 paper suggests that specialization per se results in lower extinction, not only hypercarnivory. Hence:

A- It might be the case that the lack of evidence for higher extinction rates for large hypercarnivores in the current paper (which compares it to "other Canidae"; figure 2) is, at least partially, due to the fact that highly specialized species in the other end of the spectrum (small hypocarnivores) also have higher extinction rates, and are hence "inflating" the extinction rate of "other Canidae". This could be discussed, including the possibility that breaking up the data as it was done here (hypercarnivorous vs other Canidae) might have driven the PyRate analysis shown on figure 2 which does not find

evidence for higher extinction rates in large hypercarnivores when compared to “other Canidae”. The result that hypercarnivore does not lead to higher extinction rate when compared to the other Canidae is still valid, but this is partially due to what are the “other Canidae”. Clarifying that might be helpful for the reader.

B- I wonder if the covar-model would not also be influenced by the result presented by Balisi et al 2018. In the 2018 paper the authors “find a nonlinear relationship between species duration and degree of carnivory: species at either end of the carnivory spectrum tend to have shorter durations than mesocarnivores.” Does the covar model assumes a “unidirectional” relationship between a trait value and rate? If it does, then a lack of evidence here for higher extinction rates for hypercarnivores might also be due to the fact that species at the other side of the spectrum have higher extinction rate and PyRate cannot find a slope different than 0.

If the two points above make sense, then I see two ways to deal with it. The first is simply to acknowledge in the text that those are potential methodological limitations and choices and bring the Balisi et al 2018 results into the discussion. The other is to try to re-run the analysis using three categories for the analysis shown in figure 2, and a non-linear function to relate rate to trait value for the covar model. The second option might be difficult if it is hard to implement the covar modifications, and if splitting the data into three categories diminishes the sample size to values that become too small to analyze. That said I think simply discussing might suffice for now. The paper already represents a nice advance in our understanding of canid evolution, and future work might try to deal with these methodological issues.

- These are excellent points that also have occurred to us; reading them from the reviewer is validating. We have clarified these points in the Discussion. In response to B, we opted for the first option (to acknowledge the methodological limitations in the text and highlight our previous results). The second option (to re-run the Covar analysis using three categories and a non-linear function) was not ideal for the reasons that the reviewer had provided (particularly a small sample size derived from poor fossil preservation for small hypocarnivores). As well, for our two-trait Covar analysis, to fit the current capabilities of the Covar model we had coded the dietary data as a discrete variable with three categories (hyper-, meso-, and hypocarnivory); yet this analysis generated only weak and no significant correlations between rates and traits. Therefore, for now, we have discussed these potential methodological issues and hope that future work will address them. Our additions are in lines 196–203 and lines 276–284.

2) I my previous review I pointed out that Hesperocyoninae in fact showed more evidence for more than one speciation rate (see table S2) and although the authors have mentioned that in the results it seems that the discussion still considers Hesperocyoninae to have constant speciation rate (see lines 241 to 244). A two-rate model for speciation is not only the preferred model, but if we compare a single rate model it only has 0.31 support against 0.69 for the other models which suggest 2 or more rates. To my mind this is strong evidence that your analysis does suggest

speciation is not constant through time for Hesperocyoninane. It might even have two shifts (although one shift is the preferred model). Hence the comparison to previous results might be revisited here. It seems that the results for Borophaginae is where you find strong support for differences when compared to previous work and the discussion on why this is the case is already there and a very nice one by the way.

- We agree and have modified lines 255–261 in the Discussion.

3) Although the authors refer to another paper when mentioning the estimates of prey size, it might be a good idea to briefly explain how this is done in this paper. To not break the paper flow this could be done in Figure S1 caption for example. Just a suggestion.

- We moved some of our Supplementary Figures to the Main Text in response to editorial suggestions, so the former Figure S1 is now Figure 1. We have added more detail to the relevant section in the Methods (lines 336–337).

Reviewer #3 (Remarks to the Author):

Thank you for revising your manuscript and for addressing the points raise in my previous review. I think the paper reads really nicely and provides a substantial contribution to our understanding of the evolution of carnivores. The anthropogenic impact on the extinction of large carnivores in Africa has been discussed in a recent paper (<https://onlinelibrary.wiley.com/doi/full/10.1111/ele.13451>). It's really interesting to see the temporal difference between anthropogenic effects in Africa, where humans and their ancestors have been around for much longer compared to a recently colonize continent, maybe worth a note in the paper?

- Thank you for this suggestion. Yes—the Faurby *et al.* (2020) paper is interesting and relevant to our concluding discussion regarding potential anthropogenic impacts on predator biodiversity. We have referenced the study and added a note to our paper's final paragraph, in lines 303–307.